# Multi-Path Dilated Residual Network for Nuclei Segmentation and Detection

**DOI:** 10.3390/cells8050499

**Published:** 2019-05-23

**Authors:** Eric Ke Wang, Xun Zhang, Leyun Pan, Caixia Cheng, Antonia Dimitrakopoulou-Strauss, Yueping Li, Nie Zhe

**Affiliations:** 1Harbin Institute of Technology, Shenzhen 518055, China; zhangxun@stu.hit.edu.cn; 2German Cancer Research Center, 69120 Heidelberg, Germany; c.cheng@dkfz.de (C.C.); a.dimitrakopouloustrauss@dkfz.de (A.D.-S.); 3School of Computer Engineering, Shenzhen Polytechnic, Shenzhen 518055, China; niezhe@szpt.edu.cn

**Keywords:** nuclei segmentation, microscopic pathological images observation, object detection, deep learning

## Abstract

As a typical biomedical detection task, nuclei detection has been widely used in human health management, disease diagnosis and other fields. However, the task of cell detection in microscopic images is still challenging because the nuclei are commonly small and dense with many overlapping nuclei in the images. In order to detect nuclei, the most important key step is to segment the cell targets accurately. Based on Mask RCNN model, we designed a multi-path dilated residual network, and realized a network structure to segment and detect dense small objects, and effectively solved the problem of information loss of small objects in deep neural network. The experimental results on two typical nuclear segmentation data sets show that our model has better recognition and segmentation capability for dense small targets.

## 1. Introduction

Segmentation and detection of nuclei in microscopic histopathological images is of great significance. For example, it is important to determine the morphology and location of nuclei in some cancer pathological images for grading diagnosis of malignant and benign cancers. Pathologically, clear cell renal cell carcinoma (CCRCC) [1] is the most common subtype of renal cell carcinoma. However, in clinical practice, renal carcinoma cells are often arranged in sheet, strip, acinar or tubular shape, much like renal tubules. It is very difficult to classify and diagnose them manually by pathologists. Therefore, employing computer aided image analysis technology to effectively and accurately segment the nuclei in cancer pathological images is an urgent need to analyze the malignant degree of renal cancer and build an automatic classification system for renal cancer. In addition to the nuclear segmentation of cancer pathological images, other nuclear segmentation has also value, for instance, clinical diagnosis of hematopoietic diseases can be done by observing the number, proportion and morphological changes of different types of white blood cells.

However, commonly histopathological images are highly complex, and it is difficult to segment thousands of nuclei as follows: (1) The size of high resolution histopathological images is large. (2) The type of histopathological structure is complex, and the nucleus belongs to a dense small target. There are often factors of adhesion and interference. In this kind of high-resolution image, the use of computer to automatically segment cell nuclei poses great challenges to both hardware and image analysis algorithms

## 2. Related Work

At present, active contour model [2], watershed model [3], and regional growth model [4] are three popular nuclear segmentation models. Among them, active contour model is used more frequently, mainly because it can fit the boundary of the target well. However, the segmentation effect of the model depends largely on the initial contour given by the detected nucleus position and the detection effect of the nucleus. Good or bad has a strong restriction on the final segmentation effect. Therefore, more research is needed in cell division.

In recent years, deep learning methods have been widely used in histopathological image analysis. Ciresan et al. [5] employ deep convolution neural network (CNN) to detect mitotic phenomena in mammary tissue pathological images. Ertosun et al. [6] perform automated grading of gliomas on brain histopathological images using CNN. Xu Jun et al. [7] execute automated segmentation of epithelial and matrix regions on pathological mammary tissue images using CNN. Sirinukunwattana and Ojala [8,9] use CNN to automatically segment and classify the benign and malignant glands of colorectal cancer.

In 2012, Krizhevsky A et al. [10] propose an network called AlexNet, which automatically extracts deep features of images through convolutional neural network (CNN). AlexNet uses the ReLU activation function instead of the traditional Sigmoid activation function to accelerate the convergence of the model. The powerful feature extraction ability of convolutional neural network has been rapidly applied to various fields of computer vision. Ojbect detection based on deep neural network can be divided into region extraction method and regression method. Object detection task based on region extraction method can be divided into two sub-problems: candidate region generation, candidate region boundary box regression and in-frame object recognition. In 2014, the RCNN object detection algorithm proposed by Girshick R et al. [11] is the earliest candidate region-based algorithm, which first generates candidate regions by selective search method, and then classify the candidate regions. Selective search divides the image into several smaller regions by a simple region partitioning algorithm, then merges these regions according to certain similarity rules through hierarchical grouping, and finally generates candidate regions. RCNN transforms the detection problem into classification problem. After extracting deep features from each candidate region using CNN, SVM algorithm is used to classify, which greatly improves the detection accuracy. In order to solve the problem of input fixed size image in image classification, He et al. [12] proposed SPP-Net. The proposed spatial pyramid pooling layer can be combined with RCNN to improve the performance of the model. It is extremely inefficient for RCNN to use the CNN model to compute the features of each candidate region in turn. Fast RCNN proposed by R. Girshick et al. [13] finds the corresponding feature regions from the feature map of CNN output according to the proportion of the candidate regions, which solves the problem of time-consuming repeated feature calculation. In addition, Fast RCNN uses Softmax classifier instead of SVM classifier. For Fast RCNN, it uses the selective search method in RCNN to generate candidate regions. To solve this problem, the Faster RCNN model proposed by S. Ren et al. [14] introduced candidate region generation network (RPN) to generate candidate regions directly. Faster R-CNN model has been greatly improved in both accuracy and speed of detection after RPN. The latter research based on candidate region method basically adopts similar framework. The feature pyramid network (FPN) proposed by Lin T Y et al. [15] in 2017 is one of the earliest networks to use multi-scale features and top-down structure for object detection, which improves the detection effect of small targets through multi-stage feature fusion. Li Y et al. [16] proposed the first full convolution end-to-end instance partitioning model. By introducing two score maps, the tasks of partitioning and classifying are paralleled. Mask RCNN [17] is a two-stage instance segmentation algorithm based on candidate regions. It is extended based on Faster-RCNN algorithm, and a small FCN segmentation network is added to predict the foreground and background of the target. In the first stage, the whole image is scanned to generate a number of regions that may contain objects. Each candidate region is aligned by ROI Align pooling and mapped to a fixed-size feature. Then the extracted feature is sent to the segmentation branch network, and finally the segmentation results of the instances in the candidate region are obtained. Later on, a series of object segmentation and detection schemes are proposed [18,19,20,21,22,23], however, they are not target on the high dense and small objects of the microscopic histopathological images.

## 3. Method

### 3.1. Framework

The main purpose of our scheme is realizing instance segmentation for dense small objects in histopathological images, which not only needs to find objects in the image correctly, but also needs to segment them accurately. So Instance Segmentation can be seen as a combination of object detection and semantic segmentation. For Instance Segmentation, Mask RCNN is a good framework which has better performance than state-of-the-art models. Compare with those full CNN, it can predict classes, boxes, masks of the objects at the same time, which is quite fast. However, its segmentation effect of overlapping objects is not good. So our contribution is to improve Mask RCNN by employing a designed dilated residual network. Because we find that the instance segmentation model based on dilated residual network can effectively solve the problem of small object information loss in the object detection tasks, and the model has the function of balancing the receptive field of deep network and the scale of feature map. Thus it is suitable to solve the dense small objects in histopathological images. Therefore, we use the dilated residual network as the feature extraction network of Mask RCNN model, and besides, we combine it with an improved feature pyramid structure to detect dense small objects in histopathological images. Fro comparison, we evaluate the effect of Mask RCNN framework using plain residual network, dense connection network and dilated residual network as backbone network respectively.

The flow chart of the segmentation algorithm for medical image recognition and cell nucleus recognition is shown in Figure 1, which includes multi-scale feature extraction network using D-Resnet and FPN, candidate region generation network and head network for detection and segmentation. The head network used for detection and segmentation includes three branches: sub-network of segmentation, sub-network of regression and sub-network of classification. They share the feature maps obtained by ROI Alignment Pooling, which are used to segment the foreground and background in the region of interest, regress the region boundary box, and recognize the target category in the region.

### 3.2. Region Proposal Network

Region Proposal Network(RPN) accepts images of any size as input and outputs a set of rectangular regions of interest, each candidate region has a target score. RPN uses a small network to extract candidate region features from the final feature map by sliding windows. This feature is input into two sub-full connection layers: a boundary box regression layer (reg) and a boundary box classification layer (cls). Figure 2 shows a structure of this small network.

Anchor is an important concept in RPN network. Because the target size and the ratio of length to width are different, it needs multiple scales of windows. Anchor points represent predefined candidate window sizes, and different sizes of windows are obtained according to scaling multiple and length-width ratio. For example, three multiples and three proportions can yield anchors of nine scales, as shown in Figure 2.

RPN is a multi-task model, which combines regression and classification tasks. Its cost function is Formula (Equation 1). Among them, *i* is an index of anchor frame in small batch data, and $p_{i}$ is the prediction probability of anchor frame *i* as the target. If the anchor is positive, the real label $p_{i}^{*}$ is 1, and the reverse is 0. $t_{i}$ is a vector representing the four parameterized coordinates of the predicted boundary box, and $t_{i}^{*}$ is a vector of the real boundary box related to the positive anchor frame. Classified loss $L_{cls}$ is binary logarithmic loss, and regression loss uses $Smooth_{L1}$ function. Compared with L2 loss function, L1 is insensitive to outliers or outliers, and can control the gradient to make training more convergent.
(1)$$L\left(\left\{p_{i}\right\},\left\{t_{i}\right\}\right)=\frac{1}{N_{ds}}\sum_{i}L_{cls}\left(p_{i},p_{i}^{*}\right)+\lambda \frac{1}{N_{reg}}\sum_{i}p_{i}^{*}L_{reg}\left(t_{i},t_{i}^{*}\right)$$
(2)$$Smooth_{L1}\left(x\right)=\left\{\begin{array}{c} 0.5x^{2},\left|x\right|\leq 1\\ |x|-0.5,otherwise \end{array}\right.$$
(3)$$L_{reg}=Smooth_{L1}\left(t-t^{*}\right)$$

When calculating the regression error of the boundary box, the following four coordinates are used to parameterize, where *x*, *y*, *w* and *h* represent the center coordinates and their width and height of the real boundary box. Variables *x*, $x_{a}$ and $x^{*}$ represent the predicted candidate box, Anchor, and the central abscissa of the standard box, respectively. RPN shares features to extract the final output feature graph of the network, and slides an n×n window on it. Fixed-length features are generated by RPN subnetworks and fed into the branches for classification and regression. Therefore, due to the anchor design, the boundary frames of various sizes can be predicted even if the feature has a fixed scale/proportion. RPN can be trained end-to-end through back propagation and random gradient descent. In the process of parameterization of candidate boxes, the formulas of parameterized transformation of four parameters of each candidate box are shown in Formula (Equation 4):(4)$$\begin{array}{cc} t_{x}=\left(x-x_{a}\right)/w_{a} & t_{y}=\left(y-y_{a}\right)/h_{a}\\ t_{w}=\log\left(w/w_{a}\right) & t_{h}=\log(h/h_{a})\\ t_{x}^{*}=\left(x^{*}-x_{a}\right)/w_{a} & t_{y}^{*}=\left(y^{*}-y_{a}\right)/h_{a}\\ t_{w}^{*}=\log(w^{*}/w_{a}) & t_{h}^{*}=\log(h^{*}/h_{a}) \end{array}$$

*x*, *y*, *w* and *h* represent the left side of the center point, width and height of the boundary box respectively. Variables *x*, $x_{a}$ and $x^{*}$ represent the predicted candidate box, anchor frame and the center point abscissa of the standard box respectively. Other variables are the same.

### 3.3. Feature Extraction

In the field of object detection, the main feature extraction method is multi-scale feature extraction using classical backbone network combined with feature pyramid network. Figure 1 is a typical network structure of this feature extraction method. In CNN, the receptive field refers to the size of the input region corresponding to an element in the output of a certain layer. In target detection network, because of the large receptive field of deep network, large objects use the feature prediction results of deep network output. However, the deeper the network output, the less the object edge information it contains, which makes it difficult to identify the target accurately. At the same time, the resolution of deep feature maps is often reduced to tens of times, or even smaller, and small objects are almost invisible on deep feature maps. The feature pyramid network solves this problem by using shallow feature maps. Because the semantic information of these layers is weak and the representation ability of objects is lacking, it is not easy to recognize objects correctly. As shown in Figure 3, the feature pyramid network transmits information through bottom-up and top-down paths, and at the same time adds shallow layer and deep layer with strong semantic information through horizontal connection, thus enhancing the shallow layer’s semantic expression ability. However, because small objects are already invisible in deep neural network, their semantic information in deep network would also be lost. The classical backbone network based on residual network and dense connection network originates from classification and recognition model. It only needs to predict the probability that the whole image belongs to each category. So it does not need to consider the problem that the down-sampling operation such as pooling would lead to the loss of image details. Therefore, it is necessary to increase the receptive field by multiple down-sampling. However, there would be a problem when it is directly applied to the task of object detection. That is, in order to alleviate the contradiction between the size of the receptive field and the size of the feature scale, we proposes a multi-path dilated residual network structure combined with void convolution, through which a bigger receptive field can be obtained than the plain convolution, and the feature scale does not need to be reduced.

### 3.4. Dilated Residual Network

Dilated convolution scheme, proposed by Fisher Yu et al. [24], is an improved image convolution method. Compared with plain convolution, dilated convolution has parameters of expansion rate and the size of convolution core. It increases the receptive field by changing the calculation method of standard convolution. Pooling and other down-sampling operations can also increase the receptive field, but the size of the feature map needs to be compressed, and the information compression is lossy, and the structure information of the original image can not be retained, so the dilated convolution performs better in the segmentation task that requires a lot of image position structure information. A 3 * 3 convolution core with an expansion rate of 2 is shown in the left figure of Figure 4. Dilated convolution does not increase the amount of computation and parameters, which enlarges the receptive field of convolution, so it is more conducive to extracting global features of images. How to design a reasonable deep convolution neural network based on dilated convolution is the exact problem to be solved in this paper.

Traditional convolution neural networks simply stack the pooling layers and convolution layers, for example, VGNet, which is a commonly used benchmark classification model, includes only 3 * 3 convolution core and stacks it through the basic structure of convolution, convolution and pooling. The smallest size that can obtain the eight neighborhood information of the current pixel is 3 * 3. Besides, a bigger convolution layer can be represented by multiple 3 * 3 convolutions. Multiple 3 * 3 convolution layers have more nonlinearity and fewer parameters than convolution layers of a big convolution core. At present, almost all convolution neural networks are designed with 3 * 3 convolution core. Assuming that a similar structure is used to simply stack 3 * 3 convolution cores and 2-expansion-rate dilated convolution, it can be seen that not all the pixels are involved in the computation of the network. The effect of three stacks is shown in Figure 5. This situation is a fatal problem for dense classification tasks of image segmentation at the pixel level. Therefore, in the design of network structure, we need to employ a variety of dilated convolution cores. This problem can be solved by the dilated convolution stacking with odd and even expansion rates alternately. For detection tasks, the network needs to deal with the relationship between objects of different sizes. The larger expansion rate, the greater the receptive field, the easier to deal with big objects, and the smaller expansion rate can provide more information for pixel level recognition. When the expansion rate is 1, the dilated convolution degenerates to a standard convolution, and there is no empty place. Figure 3 shows the dilated convolution with different expansion rates. Inspired by the multi-path design of Google LeNet, we design a dilated convolution structure with expansion rates of 1, 2 and 5. Considering the difficulty of training deep convolution neural networks and gradient dispersion, we design an Multi-path Dilated residual block as shown in Figure 3 based on the improvement of residual network structure to effectively solve the gradient dispersion problem, combined with multi-channel dilated convolution and 1 * 1 convolution for dimension reduction and non-linearity enhancement of feature maps.

Residual network makes the stacked non-linear multi-layer network fit another mapping relationship, then the actual mapping relationship can be expressed as. The optimization of residual mapping may be more convenient than the direct optimization of original mapping. The original residual block structure is shown in Figure 6a, and the extended residual block structure proposed in this paper is shown in Figure 6b. The structure retains two 1 * 1 convolutions in the original residual block, which are used to dimension reduced of features and ensure that the bottom feature map and the top feature map have the same dimension, so as to ensure the valid execution of the final addition operation. The shortcut (jump connection) in the dilated residual blocks combined with the leftmost path is completely equivalent to an original residual block. Besides, the dilated residual block introduces two channels of dilated convolution, which enhances the receptive field of feature maps, and fuses the multi-path features through path merging.

### 3.5. Design of Dilated Residual Network

The general deep neural network consists of 5 or 6 down-sampling operations, each down-sampling makes the length of the output feature map halved and the dimension of the feature map doubled. In this paper, we solve the problem of the conflict between the scale of feature map and the number of down-sampling by using the dilated residual block structure. In addition to the basic path of the residual network, the dilated residual block structure adds the dilated convolution path with the expansion rate of 2 and 3, and increases the receptive field of the feature map while the size of the output feature map remains unchanged. Although only the dilated convolution block cascade network is used to ensure the receptive field of deep feature, the growth of model size is unacceptable because the size of feature map keeps the same resolution as that of original image. Therefore, our DRN method ensures that the length and width of feature graph are not too big by reducing the down sampling layer in the residual network structure. Besides, the information of small objects can be effectively transmitted in the network. Figure 7 shows the design of DRN. The small network with 16 layers on the left is used to verify the feature extraction ability of the dilated residual structure in image classification tasks in subsequent experiments. The deep network structure with 64 layers on the right is used. Since the validity of the original residual network design pattern has been widely verified, the use of dilated residual blocks basically follows the design structure of the original 50-layer residual network. The 64-layer dilated residual network consists of a down sampling layer and a standard convolution layer with a convolution core of 3 and a step of 2. Through these types of network layers, the size of the output feature graph is reduced, and the number of convolution kernels is no longer increased when the number of convolution cores increase to 256, thus ensuring that the scale of the model is not too large to be used in practical applications.

In the task of marking lesions and nuclear detection in medical images, the proportion of lesions and nuclei is usually small in the whole image. Feature extraction with the original MSAR-CNN model often loses a lot of small objects information, resulting in low recognition rate of small objects. The dilated residual block structure designed in this paper can solve this problem. Similar to the combination of original residual network and characteristic pyramid structure, we design an improved Mask RCNN model based on DRN and FPN. As shown in the right figure of Figure 5, the 64-layer dilated residual network can be divided into different stages according to the depth of the output features. It naturally constitutes the bottom-up feature forward extraction path in the feature pyramid structure, while maintaining the horizontal connection and the top-down feature fusion path in FPN unchanged. It is worth noting that the feature pyramid network using DRN does not reduce the size of the feature graph in Stage 5, so Stage 5 does not include the up-sampling layer in the top-down path of the feature pyramid network. In particular, Figure 8 is based on the FPN network structure of DRN.

### 3.6. The Layer of Softmax

Softmax regression is used to solve multi-classification problems. Suppose that the training set containing m training samples is $\left\{(x^{1},y^{1}),\ldots ,\left(x^{m},y^{m}\right)\right\}$, and y∈{1,2,…,k}, for a given input *x*, it is necessary to estimate a conditional probability p=(y=j|x) for each category of the sample. Softmax can directly calculate the probability that any sample belongs to each class. Specifically, the calculation of hypothetical functions in Softmax is shown in the formula.
(5)$$p\left(i\right)=\;\frac{\exp(\theta_{i}^{T}x)}{\sum_{k=1}^{K}\exp(\theta_{k}^{T}x)}$$
(6)$$h_{\theta }\left(x^{\left(i\right)}\right)=\left[\begin{array}{c} p(y^{\left(i\right)}=1|x^{\left(i\right)};\theta )\\ p(y^{\left(i\right)}=2|x^{\left(i\right)};\theta )\\ \vdots \\ p(y^{\left(i\right)}=k|x^{\left(i\right)};\theta ) \end{array}\right]=\frac{1}{\sum_{j=1}^{k}\exp\left(\theta_{j}^{T}x^{\left(i\right)}\right)}\left[\begin{array}{c} e^{\theta_{1}^{T}x^{\left(i\right)}}\\ e^{\theta_{2}^{T}x^{\left(i\right)}}\\ \vdots \\ e^{\theta_{k}^{T}x^{\left(i\right)}} \end{array}\right]$$

In it, $\theta_{1},\theta_{2},\theta_{3},\theta_{1}k$ is the parameter of the model. The probability distribution is normalized in the form of formula, so that the sum of all probabilities is 1. Its cost function can be expressed as a formula as follows:
(7)$$[J\left(\theta \right)=\;-\;\frac{1}{m}\left[\sum_{i=1}^{m}\sum_{j=1}^{k}1\left\{y^{\left(i\right)}=j\right\}\cdot \left(\theta_{j}^{T}x^{\left(i\right)}-\log(\sum_{l=1}^{k}e^{\theta_{l}^{T}x^{\left(i\right)}})\right)\right]$$
The gradient formula obtained by using gradient descent optimization algorithm is as follows:(8)$$\begin{array}{c} \frac{\nabla J\left(\theta \right)}{\nabla \theta_{j}}=\;-\frac{1}{m}\sum_{i=1}^{m}\left[1\left\{y^{\left(i\right)}=j\right\}x^{\left(i\right)}-p\left(y^{\left(i\right)}=j|x^{\left(i\right)};\theta \right)\right] \end{array}$$

After obtaining the gradient, the gradient is brought into the gradient descent algorithm. The formula of the random gradient descent algorithm is as follows:(9)$$\theta_{j+1}\leftarrow \theta_{j}-\alpha \frac{\nabla J(\theta )}{\nabla \theta_{j}},where\;j=1,2,\ldots ,k$$

## 4. Experiments

### 4.1. Data Sets

Aiming at the problems in the process of extracting image features from the state-of-the-art backbone networks combined with feature pyramid structure, we designed an extended residual structure to alleviate the imbalance between the size of feature map and the size of sensing field. We employ two data sets: MoNuSeg and Data Science Bowl 2018 to verify the effectiveness of the structure for image feature extraction and its combination with feature pyramid structure for dense target detection and segmentation. The two datasets are described in detail as follows:

(1) MoNuSeg data set [25]

MoNuSeg contains 30 tissue microscopic images of patients, involving 7 kinds of organs and 8 kinds of diseases, and contains 16,958 labeled nuclei. The details is shown in Table 1. Tissue image of 7 kinds of organs is shown in Figure 9. The histological structure of organ tissue is mainly composed of epithelium, lumen, fat and matrix. The shape and size of various nuclei in epithelium and matrix can provide pathologists with a lot of information about whether the tissues are healthy. The image was obtained by hematoxylin-eosin staining. After staining, the nucleus become dark blue-purple, the matrix and mucus become grey-blue, and the cytoplasm becomes pink. Hematoxylin-eosin staining enhanced the comparison of nuclei with other cell structures.

(2) Data Science Bowl 2018 (DSB2018)

The data set is taken from a variety of microscopic imaging conditions, and with various of cell types, magnification and imaging modes (open field and fluorescence), is a big challenge for the generalization ability of models. Many types of images in the data set are shown in Figure 10. The data set contains 574 artificially segmented and labeled images of tissue and cell nuclei, 29,461 labeled nuclei. There is at least one nucleus in a picture. At most, there are 375 nuclei in a picture. On average, there are more than 40 nuclei in each picture.The statistical distribution of nuclei in different scale images of DSB2018 dataset is shown in Figure 11. The number of nuclei in a large number of images exceeds 25, which is a typical dense segmented dataset. The number and morphology of nuclei in each image are quite different, which brings great difficulties for detection and segmentation. In addition, the smallest nucleus in each image has only 21 pixels, while the largest one has 1037 pixels. Almost all images have overlapping cells. It is difficult to identify the nucleus even by naked eyes in some pictures.

The statistical distribution of nuclei in the images of DSB2018 data set with multiple scales is shown in Figure 11. Number of nuclei in a most images exceeds 25, which is a typical dense segmented data set.

### 4.2. Experimental Configuration and Metrics

The deep learning model experiment in this paper is based on the deep learning framework TensorFlow. Besides, in order to speed up the experimental execution, we employed Keras. Keras is a high-level deep learning framework interface written in pure Python language, which can easily work with Tensorflow. Keras can be used for simple and fast prototype design, support CNN and RNN, and also support the combination of the two networks. The parameters settings are shown in Table 2. The host configuration used in the experimental platform is shown in Table 3.

### 4.3. Evaluation Indicators

For nuclear detection and segmentation tasks in medical images, it is necessary to comprehensively evaluate the object level (nuclear recognition) and the pixel level (nuclear shape and size). F1-Score is widely used to evaluate the performance of target detection system. F1-score is defined as follows:(10)F1=2TP2TP(2TP+FP+FN)(2TP+FP+FN)
TP means the number of positive samples correctly identified as positive samples, that is, the real labeled nucleus region is correctly identified; FP means the number of negative samples misidentified as positive samples, that is, the real labeled nucleus region is misidentified as background; FN means that positive samples are misidentified as negative samples, that is, the real labeled nucleus region is misidentified as background.

Jaccard index (Jaccard similarity coefficient), can be used to compare similarities and differences between finite sample sets. The greater the Jaccard coefficient, the higher the similarity of samples.
(11)Jaccard(Bp,Bgt)=area(Bp∩Bgt)area(Bp∩Bgt)area(Bp∪Bgt)area(Bp∪Bgt)
where $B_{p}$ is the detection result location; $B_{gt}$ is the real target location; $B_{gt}$ is the overlapping area of the two areas.

We use the Aggregate Jaccard Index(AJI) to verify the effect of nuclear segmentation. The calculation method is shown in Algorithm 1. The input of the algorithm is all the real labeled nuclei in the original image and the results of all nuclei predicted by the model. For each nucleus in the original labeling, the segmentation results corresponding to the maximum Jaccard coefficient are searched, and the number of pixels is updated. For each segmentation result, the original labeling corresponding to the maximum Jaccard coefficient is searched. Finally, the intersection pixels of all matching results and the ratio of the combined pixels to the sum of the combined pixels are calculated.

**Algorithm 1** AJI computation**Require:** A series of images containing real annotations and predictive segmentation results; Gi is indexed by *i* for each real nucleus; Each prediction segmentation result Sk is indexed by *k*;**Ensure:** AJI Score 1: Initialize the number of intersections and union between the real labeled area and the predicted segmentation result area, and count them 2: C←0,U←0 3: **for all**
Gi
**do** 4:  j←argmaxk(|Gi∩Sk|/|Gi∪Sk|) 5:  Refresh the counting: C←C+|Gi∩Sj|,U←U+|Gi∪Sj| 6:  Mark Gi 7: **end for** 8: **for all**
Si
**do** 9:  j←argmaxk(|Gi∩Sk|/|Gi∪Sk|) 10:  Refresh the counting: C←C+|Gi∩Sj|,U←U+|Gi∪Sj| 11:  Mark Si 12: **end for** 13: AJI←C/U

### 4.4. Analysis

The experiment in this section aims at the nuclear detection and segmentation task in histopathological images, and verifies that the case segmentation model based on expand convolution network can effectively solve the problem of information loss of small targets in the object detection task, and that the model has the function of balancing the receptive field of deep network and the scale of feature map. We used the Multi-path Dilated residual network as the feature extraction network of Mask RCNN model, and combine it with augment feature pyramid structure deeply to detect dense small targets on histopathological images. And the effect comments of Mask RCNN model using original residual network, dense connection network and Multi-path Dilated residual network as backbone network are evaluated. Mask RCNN is a complex object detection and segmentation network. The basic algorithm flow used in the experiment is shown in Figure 1. Based on DSB2018 data set and MoNuSeg data set, two groups of experiments are designed to verify the training effect of group normalization structure and the role of D-ResNet64 backbone network in ensuring effective feature extraction and transmission in dense small object detection and segmentation network.

The image annotation method of the above data set is shown in column 4 of Figure 12. In order to use the above data set to train the segmentation model, it is necessary to use the annotation information to generate the boundary box containing the nucleus. Calculate the left-most position x1, the right-most position x2, the top position y1, and the bottom position Y2 of each labeled nucleus region in the image pixel matrix, then the boundary box of the nucleus for object detection can be uniquely determined by the left-upper vertex (x1, y1) and the right-lower vertex (x2, y2). The third column in Figure 12 is the actual extracted boundary box.

Since DSB2018 data set and MoNuSeg data set have only a few hundred training images and dozens of training images respectively, we need to propose specific data augmentation schemes to prevent model over-fitting in order to improve the generalization ability of the model. Commonly used image augmentation schemes include sharpening, Gauss noise, color image to gray level conversion, contrast and brightness adjustment, random scaling, rotation, flip, channel rearrangement. Figure 12 shows the application and effect of the above data augmentation method in data set. In the practical application process, several data augmentation methods are selected randomly to superimpose and process. At the same time, the random augmentation method is recorded to ensure that the annotated image is consistent with the original image transformation.

In actual experiment situation, because of the limitation by the physical memory of equipment, we adopt batch size 2 to train first. At the same time, because the image in the data set contains dense targets, a single image may contain hundreds of nuclei. In order to improve the recall rate of valid candidate frames in the model, it is necessary to gradually increase the maximum number of candidate region acceptable to each image. With the number of candidate regions to be processed increased, batch sizes were ultimately limited to 1. When the batch size is 1, the effect of normalized Mask RCNN using BN is significantly lower than that using GN for normalized Mask RCNN. Figure 13 shows the results of candid ate region recall and non-maximum suppression. Figure 14 shows the change of loss function value during training with GN normalization. By introducing GN, the model can still be trained stably when the batch size is 1, which provides a guarantee for further experiments.Therefore, the detection and segmentation model in this paper replaces the BN layer with the GN layer.

We evaluates the performance of two normalization methods on DSB2018 data sets with several batch sizes. The results are shown in Table 4. The training of GN has nothing to do with batch size. When the batch size is 1 or 2, the training performance of GN is consistent and the effect is the best. BN training obviously depends on batch size, and the effect decreases obviously when the batch size is 1. If the BN parameters of pre-training are used, the BN method with pre-training is much better than the BN method with small batch size in the actual training process, but it is still the best method using GN.

In order to verify that the expand residual network structure has better capability of information transmission, we designed four models, namely ResNet50 with 50-layer network structure, ResNet101 with 101-layer network structure, DenseNet121 with 121-layer network structure and D-ResNet64 with 64-layer network structure. ResNet and DenseNet are the classical backbone networks for feature extraction of detection and segmentation networks. The specific structure of each layer of the network is consistent with those described in the corresponding references. Firstly, the performance of each backbone network on DSB 2018 data set is validated. The training and testing errors of the four models on DSB 2018 data set are shown in Table 5. The error of Mask RCNN model is multi-task error, including classification error, detection error and segmentation error of each candidate region. In the training process, the training errors of the four models can converge effectively, and the test errors decrease with the increase of training rounds (Epoch), and gradually stabilize. Table 5 shows that the training errors and test errors of ReNet50 and D-ResNet64 models are close, while those of ResNet101 and DenseNet121 have bigger deviations, indicating that D-ResNet64 network has better feature extraction and generalization ability. Although DenseNet121 has the smallest training error, it does not perform as well as D-ResNet64 because of its large test error. Similarly, ResNet50 has smaller training and testing errors, but its performance of training and testing errors is lower than that of D-ResNet64. Despite the strong data augmentation method is adopted, it can be seen that ResNet101 and DenseNet121 have over-fitting phenomena with the deepening of the network because the data set itself is small.

The dilated residual network structure proposed in this paper achieves better test error on DSB2018 data set and has better capability of information transmission. We also compared the MSSD algorithm and Deep Watershed Transform algorithm for segmentation on DSB2018 data set.

The SSD method is based on the feedforward convolution network, which generates a set of bounding frames with fixed size, scores the instances of targets existing in these frames, and then suppresses the non-maxima to produce the final detection results. Mask SSD is implemented by adding Mask branches to the SSD network. Figure 15 and Figure 16 and Table 6 are the prediction results of Mask SSD based on ResNet50 network on DSB2018 data set.

In order to further verify the advantage of D-ResNet64 as the backbone network of Mask RCNN in dense small object detection, this paper uses ResNet101 and D-ResNet64 as the control group and experimental group, and carries out experiments on MoNuSeg data set. The evaluation results of ResNet101 and D-ResNet64 in MoNuSeg dataset are shown in Table 2. According to the performance of the two models on DSB2018 data, D-ResNet64 should be better than ResNet101 in F1-Score and AJI indicators. The actual evaluation results on the MoNuSeg data set are shown in Table 7, and images of different organs are selected as the test set to evaluate the effectiveness of the algorithm.

Further, by comparing and analyzing the results of the two algorithms, we selected the more representative images on the MoNuSeg data set for the reflect group and the test group to obtain the segmentation results, respectively, as shown in Figure 17. By observing the experimental results of 18 samples, it can be clearly found that the D-ResNet64 Mask RCNN used in this paper is superior to the classical ResNet101 model in dense small object detection tasks and has a higher detection rate in the area marked by blue box.

In order to achieve a clear understanding of the detection and segmentation effect of each algorithm on various organ cells, Table 7 is plotted as a histogram as shown in Figure 18. It can be seen that Mask RCNN based on D-ResNet64 performs better in both indicators.

Concluding the experimental results, it can be seen that the dilated residual network for the object detection of dense small targets proposed in this paper has a better performance of feature extraction for detection and segmentation tasks.

## 5. Conclusions

We mainly designed a model of the dilated residual network structure based on multi-channel empty convolution for nuclei segmentation and detection. The validity of the dilated residual network structure is verified on the medical image data sets. We find that the performance of 64-layer dilated residual network is obviously better than that of 50-layer original residual network, and it has the performance of close to 101-layer original residual network. Based on Mask RCNN model, a network structure for dense small object detection and segmentation is designed and experimented. The network detection and segmentation capabilities under group normalization and batch normalization are compared respectively. Experiments show that the effect of group normalization is better than that of batch normalization when batch size is small. In addition, we compared the performance in detail for employing dilated residual network as backbone network and other classical backbone networks. Experiments show that the former has a better performance in the object detection of small nuclei targets. The dilated residual network structure proposed in this paper can enhance the function of convolution module. Compared with the original model, the model using dilated residual network structure has a better performance in the detection and segmentation of dense small targets. 

## Figures and Tables

**Figure 1 cells-08-00499-f001:**
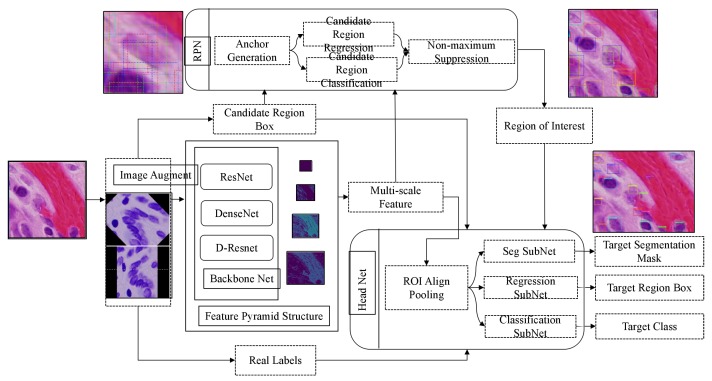
The Process of Improved Algorithm of Nuclei Dense Detection and Segmentation.

**Figure 2 cells-08-00499-f002:**
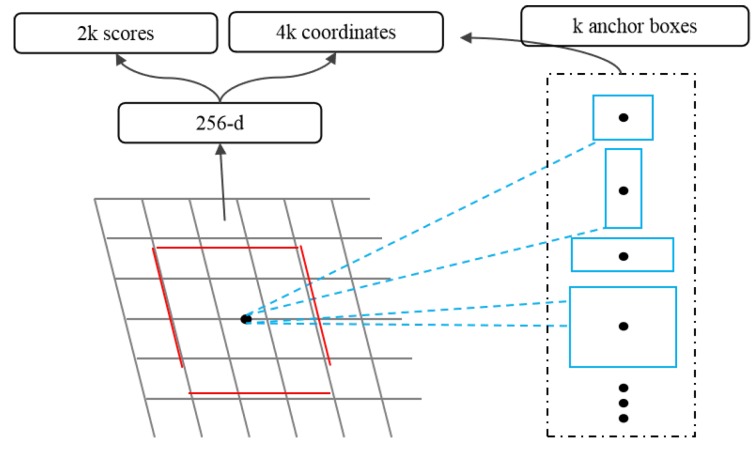
Candidate Region Generation Network Architecture.

**Figure 3 cells-08-00499-f003:**
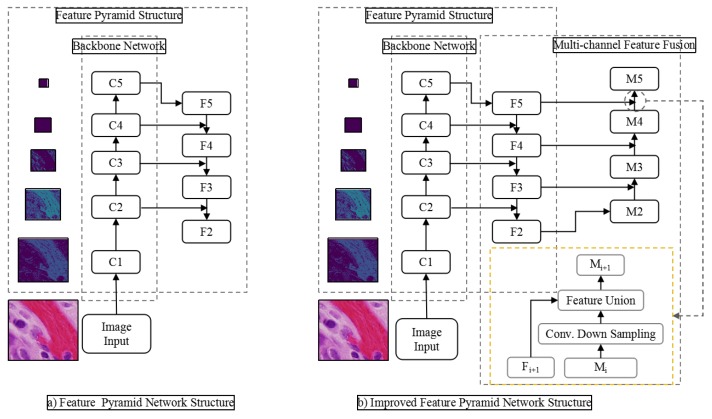
Multi-scale feature extraction structure diagram of backbone network combined with feature pyramid network.

**Figure 4 cells-08-00499-f004:**
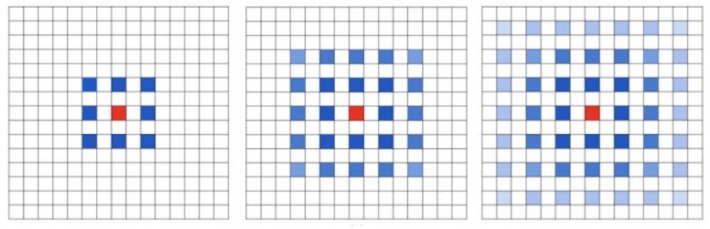
3 layers stack up with 3 * 3 convolution core.

**Figure 5 cells-08-00499-f005:**
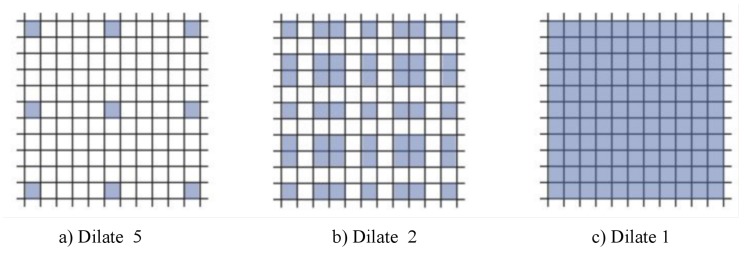
The dilated convolution of the convolution core is 3 × 3, and the expansion rates are 5, 2 and 1, respectively.

**Figure 6 cells-08-00499-f006:**
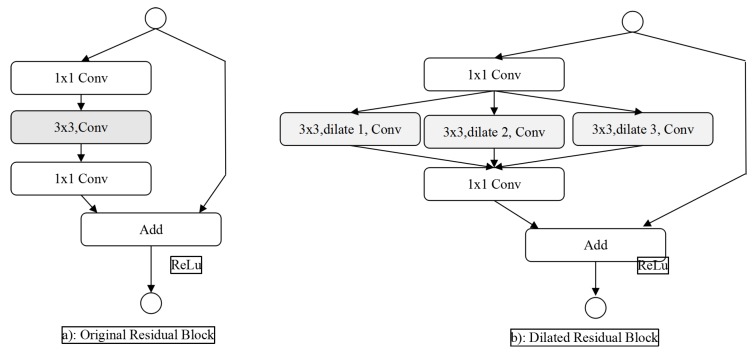
Residual Network and Dilated Residual Network.

**Figure 7 cells-08-00499-f007:**
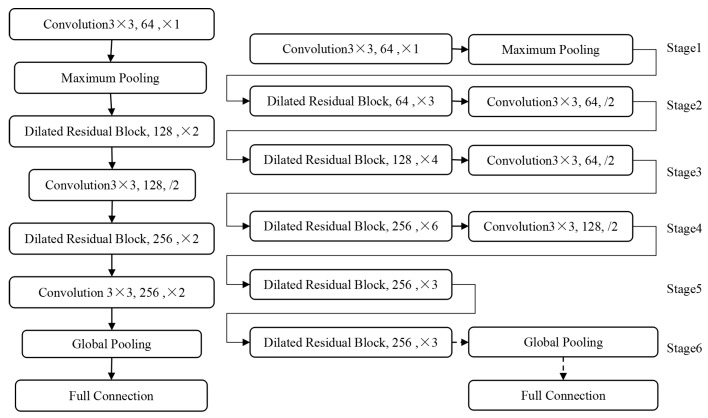
Deep Residual Network, Left: 16 layers, Right: 64 layers.

**Figure 8 cells-08-00499-f008:**
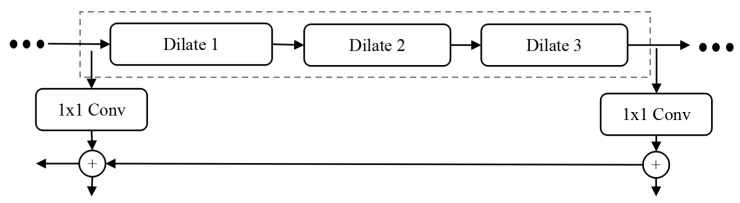
FPN Network based on DRN.

**Figure 9 cells-08-00499-f009:**
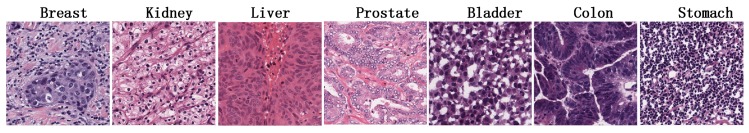
Data Set MoNuSeg.

**Figure 10 cells-08-00499-f010:**
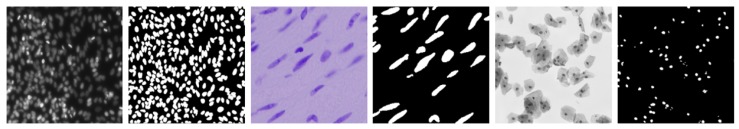
Data Set DSB2018.

**Figure 11 cells-08-00499-f011:**
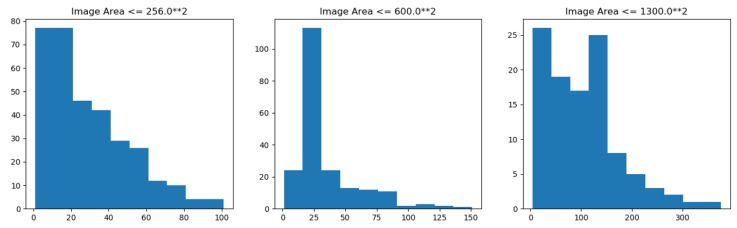
Statistical Distribution of Nucleus Number in Different Size Images (Left: 256, Center: 600, Right: 1300).

**Figure 12 cells-08-00499-f012:**
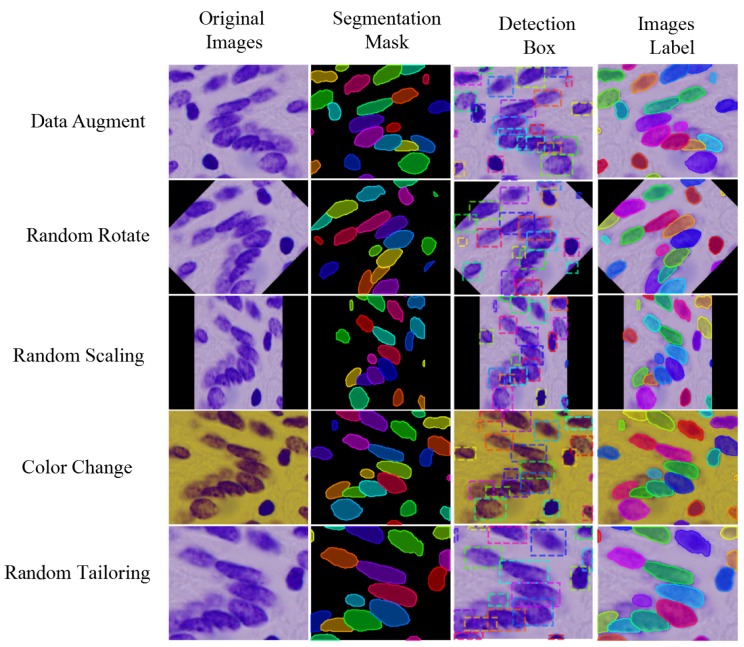
Data Augmentation Method and its Effect.

**Figure 13 cells-08-00499-f013:**
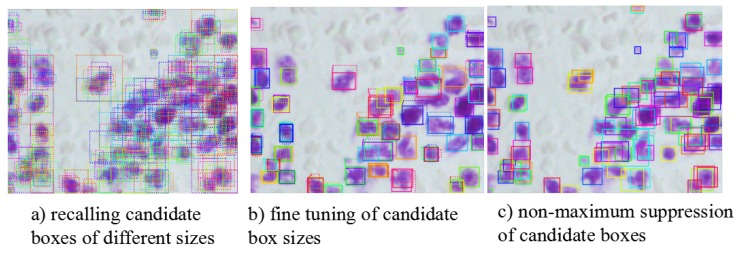
Candidate Region Processing in Nuclear Detection.

**Figure 14 cells-08-00499-f014:**
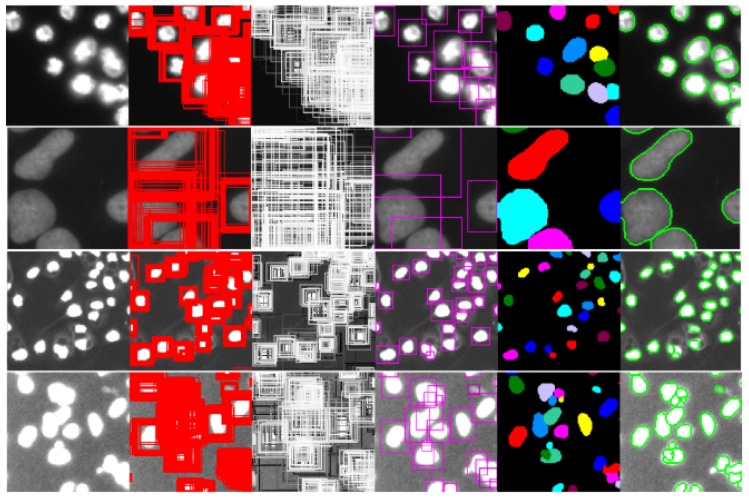
Prediction Result of ResNet50 MASK SSD on DSB2018.

**Figure 15 cells-08-00499-f015:**
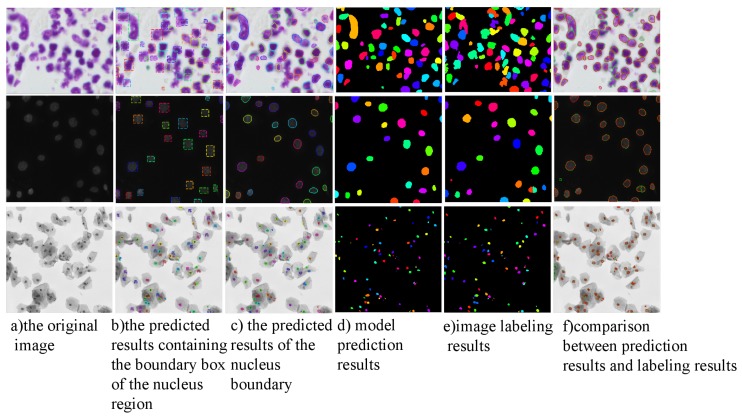
Comparison of the methods on DSB2018.

**Figure 16 cells-08-00499-f016:**
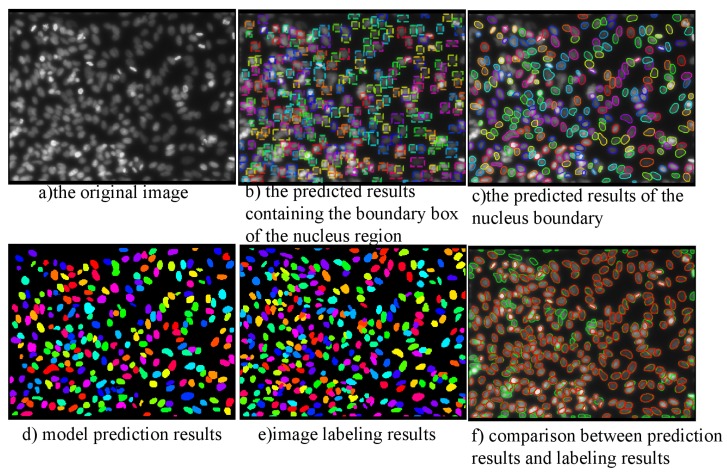
Prediction Result of D-ResNet64 Mask RCNN on DSB2018.

**Figure 17 cells-08-00499-f017:**
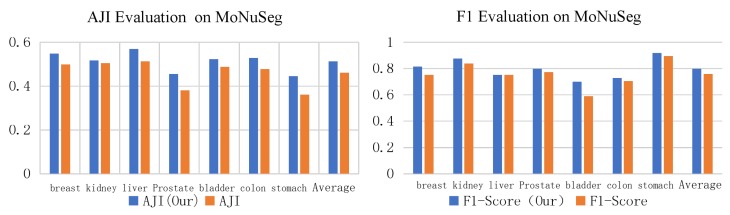
Segmentation Results of Dense Small Target Detection and Segmentation Model in MoNuSeg Data Set.

**Figure 18 cells-08-00499-f018:**
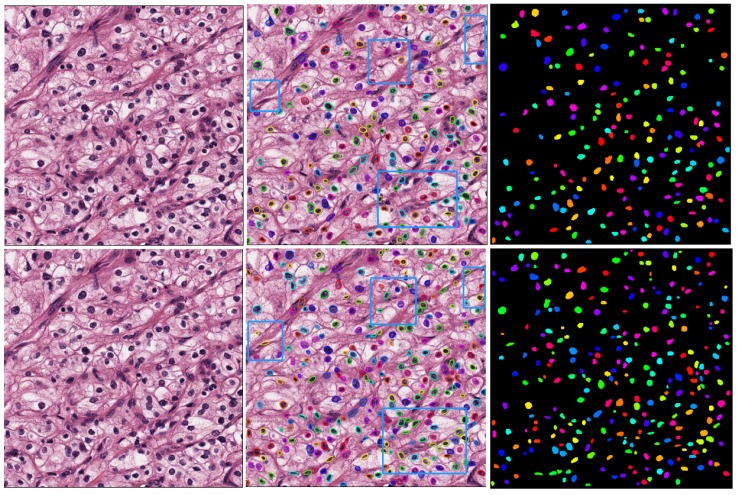
Segmentation Results of Dense Small Target Detection and Segmentation Model in Monseg 2018 Data Set.

**Table 1 cells-08-00499-t001:** The Statistic Details of MonuSeg.

Ogan	Desease	Number of Images (piece)	Mark of Nuclei (number)
Breast	Invasive breast cancer	6	2213
Kidney	Clear cell carcinoma of kidney	6	5577
Lung	Squamous cell carcinoma of lung, Lung adenocarcinoma	6	2743
Prostate	Prostate adenocarcinoma	6	2400
Bladder	Bladder urothelial carcinoma	2	743
Colon	Colonic adenocarcinoma	2	726
Stomach	Gastric adenocarcinoma	2	2556
Total	-	30	16,958

**Table 2 cells-08-00499-t002:** Deep Learning Parameters Setting.

Parameter	Value
Learning Rate	0.0001, 0.00001, 0.000001
Epochs	20, 40, 75
Optimization	amsgrad
GradientClipNorm	5.0

**Table 3 cells-08-00499-t003:** Statistic Details of MonuSeg.

Project	Configuration
OS	Linux | Ubuntu 16.04 LTS
CPU	Intel Core i7-6800K @ 3.40 GHz
GPU	2*NVIDIA GeForce GTX 1080 (8G Graphic Memory)
Memory	48G Mem + 64G Swap
Harddisk	2T
Development Language	Python 3
Deep Learning Framework	TensorFlow, Keras, Pytorch

**Table 4 cells-08-00499-t004:** AJI Score by BN and GN with Data Science Bowl.

Normalization Method	BN (Training)	BN (No Training)	GN (Training)
Batch Size 1	0.3442	0.4191	0.4514
Batch Size 2	0.3707	0.4243	0.4527

**Table 5 cells-08-00499-t005:** Training Errors and Test Errors of Models on DSB2018 Data Set.

Backbone Net	ResNet50	ResNet101	DenseNet121	D-ResNet64
Epoch	20	25	25	30
Training error	1.83	1.08	0.91	1.25
Test error	2.05	1.74	1.56	1.37

**Table 6 cells-08-00499-t006:** Results of comparison models on DSB2018 data set.

Model	AJI (Gray)	AJI (ALL)
SVM	0.0656	0.0653
Random Forest	0.1716	0.1247
Logistic Regression (LR)	0.4815	0.3444
UNet + Morphology Postprocessing	0.4785	0.3647
ResNet50 + Mask RCNN	0.5560	0.4527
ResNet101 + Mask RCNN	0.6094	0.5155
DenseNet121 + Mask RCNN	0.5681	0.4791
ResNet50 + Mask SSD[34]	0.5819	0.4763
UNet + Deep Watershed Transform[57]	0.6308	0.5016
D-ResNet + Mask RCNN (Our)	0.6077	0.5249
D-ResNet + Mask RCNN + PA(Our)	0.6145	0.5440

**Table 7 cells-08-00499-t007:** Comparison of detection and segmentation effects between the improved model and the original model on MonuSeg dataset.

Organ	AJI (Ours)	AJI (ALL)	F1-Score (Our)	F1-Score
Breast	0.5486	0.4993	0.8167	0.7532
Kidney	0.5175	0.5051	0.8768	0.8386
Liver	0.5692	0.5137	0.7532	0.7524
Prostate	0.4561	0.3804	0.7984	0.7731
Bladder	0.5234	0.4876	0.7009	0.5894
Colon	0.5291	0.4783	0.7293	0.7057
Stomach	0.4456	0.3618	0.9187	0.8962
Average	0.5128	0.4609	0.7991	0.7584

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
