# Peer review of "Multi-Path Dilated Residual Network for Nuclei Segmentation and Detection"

_cells, 2019, doi:10.3390/cells8050499_

Round 1

Reviewer 1 Report

In this paper, the authors proposed a multi-path dilated residual network for Nuclei segmentation and detection, which is based on Mask RCNN. Moreover, the developed a network structure to segment and detect small objects, in order to solve the problem of information loss of small objects in a deep neural network. In fact, the proposed approach has been validated on two typical nuclear segmentation data sets: MoNuSeg and Data Science Bowl.

Generally, the presented idea in this manuscript is interesting.

However, major revisions have to be made and some parts of this manuscript are not complete to claim the advantages of the proposed method.

There are some suggestions that would increase the strength of the paper which is listed bellows:

1)    One of my major points about your article concerns the consistency of this analysis. Authors mentioned many exciting topics and strive to connect them. I would suggest the authors provide a better connection between these algorithms.

2)    Moreover, the proposed approach is not clear. I would like that the authors add a formal framework, for example, an algorithm or mathematical model.

3)    In experimental results, for the training, could the authors add some details about the learning and parameters settings, i.e., learning rate, epochs, optimization, and regularization?

4)    The English and format of this manuscript should be checked very carefully.

Author Response

Dear Reviewer,

        Thanks for your comments. We submit the revision of our manuscript. And please check the attachment file for the response to reviewer.

        Best Regards,

Eric Ke Wang

Reviewer 2 Report

This paper proposes a multi-path dilated residual network based on improved deep learning methods to segment and detect nuclei in the histopathological images. The authors claim that experimental results demonstrate their superiority of their model on nuclei segmentation and detection and also show that the dilated residual network structure is better at detecting small nuclei targets compared to so-called classical backbone networks. This paper is interesting and the authors elaborately describe the method. The results also seem to be promising. However, I have some concerns which require the authors to address:

(1) Although this paper compares different models like so-called dilated residual network and classical backbone networks, I did not see any comparison of the proposed method against state-of-the-art methods like SPP-Net, fast RCNN, etc, as mentioned by the authors. Besides, although deep learning has been demonstrated to outperform traditional machine learning methods like SVM [1-2], logistic regression [3-4], etc, it would still be interesting to see the performance comparison between the proposed method against those traditional methods, because these traditional methods are still prevalently used in biomedical community [1-4].

[1] Zhang, X., & Liu, S. (2016). RBPPred: predicting RNA-binding proteins from sequence using SVM. Bioinformatics33(6), 854-862.

[2] S. Wan, M. W. Mak, and S. Y. Kung, "Mem-ADSVM: A Two-Layer Multi-Label Predictor for Identifying Multi-Functional Types of Membrane Proteins", Journal of Theoretical Biology, 2016, vol. 398, pp. 32-42.

[3] Algamal, Z. Y., & Lee, M. H. (2015). Penalized logistic regression with the adaptive LASSO for gene selection in high-dimensional cancer classification. Expert Systems with Applications42(23), 9326-9332.

[4] S. Wan, M. W. Mak, and S. Y. Kung, " mPLR-Loc: An Adaptive-decision Multi-label Classifier Based on Penalized Logistic Regression for Protein Subcellular Localization Prediction", Analytical Biochemistry, 2015, vol. 473, pp. 14-27.

(2) Although the authors spent a large section to describe their proposed method, it is still unclear how the method is logically structured. For example, Fig. 7, seemingly the flowchart of the method, is not clearly presented. Are their two inputs (the two images on the left side) for the whole method? What are the relationships between each big block? What are the relationships between each small block (e.g., ResNet, DenseNet, D-Resnet) within a big block? Do you run each small block for each dataset, or just select some? Besides, some texts (e.g., in the top block) are messily presented in the flowchart. Furthermore, what are the relationships, between Fig.1 until Fig. 6?

(3) What is AJI score/review used in Tables 3-6?

(4) References [11-13] in the manuscript were not cited properly in the introduction part.

Author Response

(The authors gave the same response as above.)

Round 2

Reviewer 1 Report

The manuscript has improved the quality of the paper significantly. I recommend to accept the manuscript for the publication.

Author Response

Dear Sir,

Thanks again!

Best Regards,

Eric Ke Wang

Reviewer 2 Report

The current version has been greatly improved compared to the original one. However, the authors have ignored to address one issue in Comment 1 in the last round of review, i.e., comparing their method with logistic regression mentioned in Refs [1-4] in my previous comments.

Author Response

Dear Sir,

         Thanks for your comments. Please see the attachment for the response to the reviewer. 

         Thanks again.

Best Regards,

Eric Ke Wang